# Feasibility study of a multicentre cluster randomised control trial to investigate the clinical and cost-effectiveness of a structured diagnostic pathway in primary care for chronic breathlessness: protocol paper

Gillian Doe ,[1] Jill Clanchy,[2] Simon Wathall,[3] Stacey Chantrell,[4] Sarah Edwards,[4] Noel Baxter,[5] Darren Jackson,[6] Natalie Armstrong,[7] Michael Steiner,[1,4] Rachael A Evans [1,4]

For numbered affiliations see end of article.

**Correspondence to**
Dr Rachael A Evans;
re66@leicester.ac.uk

## ABSTRACT

**Introduction** Chronic breathlessness is a common and debilitating symptom, associated with high healthcare use and reduced quality of life. Challenges and delays in diagnosis for people with chronic breathlessness frequently occur, leading to delayed access to therapies. The overarching hypothesis is a symptom-based approach to diagnosis in primary care would lead to earlier diagnosis, and therefore earlier treatment and improved longer-term outcomes including health-related quality of life. This study aims to establish the feasibility of a multicentre cluster randomised controlled trial to assess the clinical and cost-effectiveness of a structured diagnostic pathway for breathlessness in primary care.

**Methods and analysis** Ten general practitioner (GP) practices across Leicester and Leicestershire will be cluster randomised to either a structured diagnostic pathway (intervention) or usual care. The structured diagnostic pathway includes a panel of investigations within 1 month. Usual care will proceed with patient care as per normal practice. Eligibility criteria include patients presenting with chronic breathlessness for the first time, who are over 40 years old and without a pre-existing diagnosis for their symptoms. An electronic template triggered at the point of consultation with the GP will aid opportunistic recruitment in primary care. The primary outcome for this feasibility study is recruitment rate. Secondary outcome measures, including time to diagnosis, will be collected to help inform outcomes for the future trial and to assess the impact of an earlier diagnosis. These will include symptoms, health-related quality of life, exercise capacity, measures of frailty, physical activity and healthcare utilisation. The study will include nested qualitative interviews with patients and healthcare staff to understand the feasibility outcomes, explore what is 'usual care' and the study experience.

**Ethics and dissemination** The Research Ethics Committee Nottingham 1 has provided ethical approval for this research study (REC Reference: 19/EM/0201). Results from the study will be disseminated by presentations

> ## Strengths and limitations of this study
>
> ► A mixed methods approach will be used to both understand how breathlessness is perceived and diagnosed in primary care, and further interpret the findings of the feasibility study.
> ► The structured diagnostic pathway uses existing investigations available in primary care which will aid future implementation.
> ► The trial is embedded within clinical care and uses opportunistic recruitment when patients present with breathlessness in primary care.
> ► The study design is a cluster randomised trial to minimise potential bias and contamination.
> ► The trial will be conducted in a single region in the UK and may limit the generalisability of the study findings.

at relevant meetings and conferences including British Thoracic Society and Primary Care Respiratory Society, as well as by peer-reviewed publications and through patient presentations and newsletters to patients, where available.
**Trial registration number** ISRCTN14483247.

## INTRODUCTION

Breathlessness is associated with high healthcare use, accounting for 5% of presentations to the emergency department,[1 2] approximately 4% of general practitioner (GP) consultations[3] and reported by patients in 12% of medical admissions.[4] Breathlessness is reported by around 9%–11%[5 6] of the general population, varying with severity, socioeconomic status[6 7] and increasing with age to 25% in people over 70 years old.[8 9] Functional impairment from breathlessness, measured using the Medical Research Council (MRC)

dyspnoea scale, is associated with reduced survival regardless of underlying diagnosis.[10]

Two-thirds of breathlessness is caused by cardiorespiratory conditions.[11] Clinical data shows that for patients over the age of 40 the most common causes of breathlessness are chronic obstructive pulmonary disease (COPD), heart failure (HF), obesity, anaemia and anxiety.[12 13] These conditions can be potentially diagnosed or excluded using investigations that can be performed in community settings.[12] All of these investigations are recommended in the National Institute for Health and Care Excellence (NICE) clinical knowledge summary[14] but there is no specific guidance on the timeframe within which they should be performed.

Primary care data have highlighted many missed opportunities over many years to diagnose conditions associated with breathlessness, such as COPD and HF,[15 16] with many patients being diagnosed only when the disease is severe or requiring hospitalisation.[15 16] These data indicate significant challenges in the deployment of simple diagnostics in the primary care setting. There are also well-documented misdiagnoses for COPD, asthma and interstitial lung disease across healthcare settings.[15 17–21]

Our overarching hypothesis is that a symptom-based approach for diagnosis in primary care for patients with chronic breathlessness will lead to earlier diagnosis, earlier treatment and improved outcomes such as health-related quality of life (HRQoL). However, it is also important to consider the implications of over investigating and 'over-diagnosis' in patients and find the balance between clinical and cost-effectiveness for a diagnostic pathway.[22] A large and potentially expensive multicentre cluster randomised controlled trial (RCT) would be necessary to understand the clinical and cost-effectiveness of a structured diagnostic pathway for chronic breathlessness. The scope of this study is to assess the feasibility of such a trial and help inform the design.

For this feasibility study the specific aims are:

1. To assess feasibility by recruitment and retention rate of patients in the trial to enable calculation of the

| Table 1 | Secondary outcome measures | |
|---|---|
| **Secondary outcome measures** | **Measurement tool** |
| Proportion of diagnoses in usual care and intervention within 1 year of presentation | Review of healthcare records for all participants |
| Time to diagnosis | Review of healthcare records for symptom presentation and diagnosis date |
| Health-related quality of life | Chronic Heart Questionnaire EuroQol 5 Dimensions 5 Levels |
| Breathlessness | Dyspnoea–12 Multidimensional Dyspnoea Profile Medical Research Council dyspnoea scale Baseline Dyspnoea Index and Transition Dyspnoea Index |
| Physical activity | Activity monitors (GENEActiv and ActiGraph devices) to measure daily step count, sedentary time, moderate and vigorous activity |
| Exercise capacity | Incremental Shuttle Walk Test |
| Frailty | Short Performance Physical Battery, Fried's frailty score, Rockwood frailty score, handgrip and quadriceps strength. |

number of GP practices, cluster sizes and duration of the ultimate RCT (box 1).

2. To better understand 'usual care' through prospective observation and qualitative analysis, and to understand any influence of the trial design on usual care.

3. To determine the proposed primary outcome measure for the future trial and to increase understanding of what is an important and realistic difference while exploring potential of other outcome measures (table 1).

4. Identify sources of data and how best to collect these in order to plan the economic evaluation that would accompany a full trial.

## METHODS AND ANALYSIS

This is a mixed methods study designed using the MRC guidelines on developing complex interventions.[23]

### Trial design and registration

This is a 1-year feasibility cluster RCT recruiting from primary care. Ten GP practices from East and West Leicestershire and Leicester City clinical commissioning groups (CCG) will be cluster randomised to a structured diagnostic pathway or usual care. The intervention practices will follow a structured diagnostic pathway to include early investigations. Usual care will continue without any intervention.

---

**Box 1  Feasibility measures**

Feasibility measures

► Number of patients recruited per week per general practitioner (GP) practice population size.
► Number of participating GP practices verses the number approached.
► Time for GPs to screen for eligibility.
► Number of eligible patients who agree to be approached by the research team verses total number of eligible patients.
► Number and timing of investigations in the diagnostic pathway completed.
► Acceptability of the research visit to the participants.
► Data collected from interviews regarding participant experience of the trial.
► Data collected from interviews regarding GP experience of participating in the trial and influence on their practice.

The University of Leicester will act as study sponsor and the trial has been registered on the ISRCTN website.

## Patient and public involvement

Prior to the trial design an engagement event was held with clinicians and patients from relevant services, including GPs, community and hospital clinicians with cardiorespiratory background and patients with experience of chronic breathlessness to discuss the optimal breathlessness pathway using Listening into Action.[24] The structured pathway to be used in the trial was the output from this engagement, the NICE guidance and the Breathlessness IMPRESS Tips for Clinicians guidance.[12 14]

The National Institute for Health Research Biomedical Research Centre patient and public involvement (PPI) groups were also consulted about the study design including the duration of the research visits, patient facing information and questionnaire packs. They provided feedback on the type and ordering of the questionnaires and ways to reduce burden to patients. The wording for the electronic template to aid recruitment was developed by members of the PPI group. The trial management and steering groups will have patient members. The study team aim to feed back to all local PPI partners with results from this trial.

## Participants

### Eligibility criteria for patients

Patients will be eligible if they are over 40 years old, experienced breathlessness for over 2 months and are within their first two presentations to primary care with symptoms of breathlessness. Exclusion criteria are an existing diagnosis for their current symptom of breathlessness, an estimated prognosis of less than 1 year or if the patient requires immediate hospitalisation for their symptoms.

### Eligibility criteria for GP practices

GP practices will be approached to take part in this study if they serve a patient population over 10 000. Practices that are research active, as identified by the local Clinical Research Network research scheme, will be approached. The practices will be visited by the study team to discuss taking part in the study and engage with the practice teams.

The GP practices will be randomised 1:1 and stratified by CCG using Statarand,[25] a Stata randomisation module (Boston College Department of Economics).

## Recruitment

Patients will be recruited over 1 year. Patients who meet the above eligibility criteria will be approached in primary care when they present with symptoms of chronic breathlessness. An electronic template on the patient record, triggered at the point of consultation, will be used to aid opportunistic recruitment (see figure 1). The template will be triggered by either free text or Read codes relating to breathlessness. Limits have been set on the trigger to avoid it appearing for patients who have an existing diagnosis of COPD or HF. The template summarises the study, prompting the GP to ask if the patient gives consent to be contacted by the study team. The GP will select yes or no as appropriate to the patient agreeing to have their contact details sent to the study team.

The electronic template has been developed in partnership with Keele Clinical Trials Unit who will support implementation onto the electronic patient record system (SystmOne and EMIS) for each practice. This approach has been used successfully in other primary care trials.[26] The Leicester City CCG who take the research lead for Leicester and Leicestershire were consulted regarding the use of the electronic template.

## Setting

The GP practices are located in Leicester and Leicestershire, England, UK. The research team is based at Glenfield Hospital, University Hospitals of Leicester NHS Trust. Participants will be invited to attend Glenfield Hospital for a research visit.

## Safety reporting

Participation is considered to be low risk. It is believed the occurrence of any serious adverse events (SAEs) will be low. Participants will be undertaking some physical tests as part of the research. There may be a small risk of

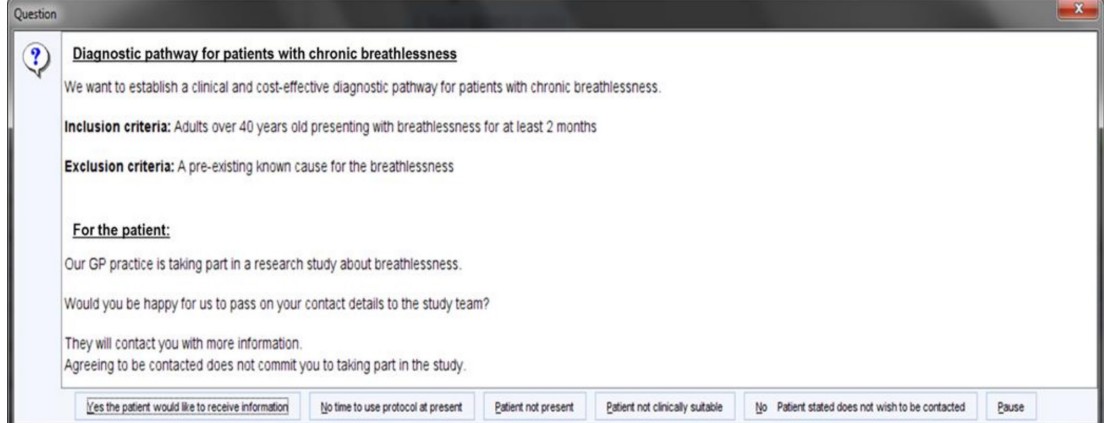

**Figure 1** The electronic template triggered on patient electronic healthcare record. GP, general practitioner.

worsening breathlessness, changes in blood pressure and changes in heart rate and a very small risk of falls. Trained staff and emergency equipment will be available to deal with any serious events. All adverse events and SAEs will be recorded on the adverse event log. Any SAEs related to the physical tests conducted as part of the study will be reported.

### Procedure

A weekly search will be performed by practice staff to provide a report of patients who have agreed to be contacted by the study team following discussion with the GP, as described above. The report will be sent via secure encrypted nhs.net email and patients will be contacted to complete telephone screening for eligibility. A script for telephone screening will be available for use to confirm patients' age, previous medical history and to explain the study in more detail. If patients agree to hear more about the study they will be sent a participant information sheet

by post and, where willing, a provisional appointment letter. When patients attend their research visit, they will complete written informed consent (online supplemental file 1) and their GP will be notified of their involvement in the study. For patients who decline to take part in the study or are ineligible, the reasons will be documented and collated.

Where possible, patients will attend their research visit within 1 month of seeing the GP with their breathlessness symptoms. Patients will attend a second research visit 12 months after their initial appointment and also be contacted by phone at 6 months and asked to complete the questionnaire pack, which will be sent in the post. Please see figure 2 for the study schedule.

### Intervention—structured diagnostic pathway

Patients who attend GP surgeries in the intervention group will undergo a set of investigations within 1 month; body mass index (BMI), spirometry, ECG, chest X-ray,

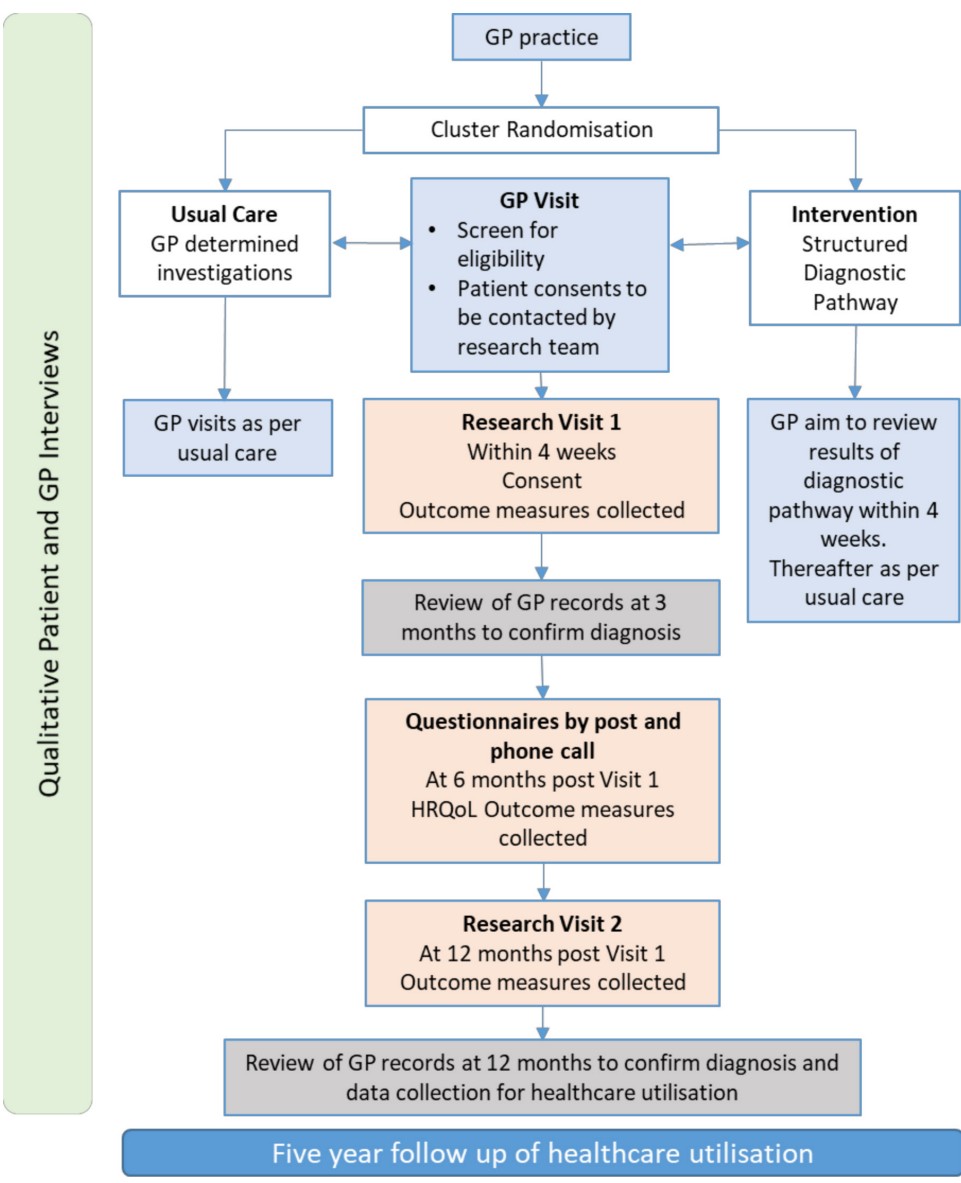

**Figure 2** Study schedule. GP, general practitioner; HRQoL, health-related quality of life.

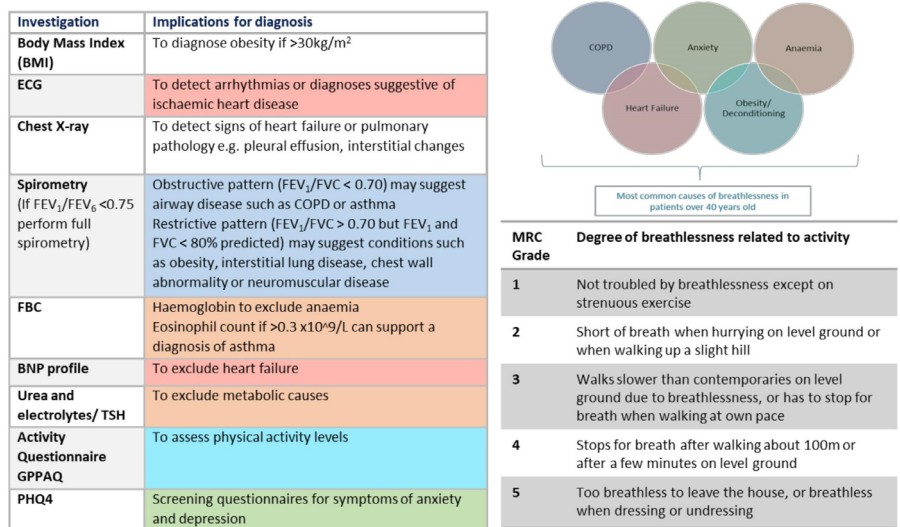

**Diagnostic pathway for initial presentation of chronic breathlessness**

For adults > 40 yrs old presenting with breathlessness for > 2 months please follow this pathway.

Please Read code for Breathlessness. For acute breathlessness, please follow usual procedure for assessment and action.

History → Examination → Investigations → Review

**STEP 1. Focused History and Examination**

As usual, please take a clear history to assess possible causes and impact of breathlessness. Please include the following:

| HISTORY | EXAMINATION |
|---|---|
| ✓ Onset and duration of breathlessness | ✓ Vital signs: resting HR and rhythm, $O_2$ saturation, RR and BP |
| ✓ At rest/exertional | ✓ Observe general appearance and breathing pattern (increase use of accessory muscles) |
| ✓ Nature of breathlessness | ✓ Assess JVP |
| ✓ Aggravating and relieving factors | ✓ Check for peripheral oedema |
| ✓ Associated symptoms (e.g. chest pain, cough, wheeze, ankle swelling, palpitations) | ✓ Auscultate lungs (particularly for bi-basal crackles) |
| ✓ Orthopnoea, PND | ✓ Auscultate cardiac sounds (listen for murmurs including right carotid area for aortic stenosis) |
| ✓ Levels of exercise and daily activity | ✓ BMI (weight kg/height $m^2$) |
| ✓ Impact on everyday life and MRC Dyspnoea scale | |
| ✓ Co-morbidities and medications | |
| ✓ Smoking history including pack years and substance smoked | |
| ✓ Environmental and occupational risk factors | |

**STEP 2. Investigations for Chronic Breathlessness**

Please initiate the investigations listed below to rule in/out common causes of breathlessness.

| Investigation | Implications for diagnosis |
|---|---|
| Body Mass Index (BMI) | To diagnose obesity if >30kg/$m^2$ |
| ECG | To detect arrhythmias or diagnoses suggestive of ischaemic heart disease |
| Chest X-ray | To detect signs of heart failure or pulmonary pathology e.g. pleural effusion, interstitial changes |
| Spirometry (If $FEV_1/FEV_6$ <0.75 perform full spirometry) | Obstructive pattern ($FEV_1/FVC < 0.70$) may suggest airway disease such as COPD or asthma. Restrictive pattern ($FEV_1/FVC > 0.70$ but $FEV_1$ and FVC < 80% predicted) may suggest conditions such as obesity, interstitial lung disease, chest wall abnormality or neuromuscular disease |
| FBC | Haemoglobin to exclude anaemia. Eosinophil count if >0.3 x$10^9$/L can support a diagnosis of asthma |
| BNP profile | To exclude heart failure |
| Urea and electrolytes/ TSH | To exclude metabolic causes |
| Activity Questionnaire GPPAQ | To assess physical activity levels |
| PHQ4 | Screening questionnaires for symptoms of anxiety and depression |

Most common causes of breathlessness in patients over 40 years old

| MRC Grade | Degree of breathlessness related to activity |
|---|---|
| 1 | Not troubled by breathlessness except on strenuous exercise |
| 2 | Short of breath when hurrying on level ground or when walking up a slight hill |
| 3 | Walks slower than contemporaries on level ground due to breathlessness, or has to stop for breath when walking at own pace |
| 4 | Stops for breath after walking about 100m or after a few minutes on level ground |
| 5 | Too breathless to leave the house, or breathless when dressing or undressing |

**STEP 3. Please aim to review results and patient within 1 month.** Please use additional document for guidelines and onward referral as appropriate

Diagnostic Uncertainty → Please consider referral to Breathlessness Clinic (referral form in PRISM)

**Figure 3** Diagnostic pathway. BNP, brain natriuretic peptide; BP, blood pressure; COPD, chronic obstructive pulmonary disease; FBC, full blood count; FEV1, forced expiratory volume in 1 s; FVC, forced vital capacity; GPPAQ, General Practice Physical Activity Questionnaire; HR, heart rate; JVP, jugular venous pressure; MRC, Medical Research Council; PHQ-4, Patient Health Questionnaire 4; PND, paroxysmal nocturnal dyspnoea; PRISM, Patient and Referral Implementation SysteM; RR, respiratory rate; TSH, thyroid stimulating hormone.

full blood count, N-terminal (NT)-pro hormone brain natriuretic peptide (NT-proBNP) profile, anxiety and depression screening using the Patient Health Questionnaire—4 item (PHQ-4)[27] and the General Practice Physical Activity Questionnaire.[28] The GPs and clinicians will be provided with the pathway document to support a structured history, examination and use of the investigations (figure 3). The electronic template will prompt the GP to action these investigations. The pathway will also

be provided as a laminated document for each clinician in the intervention practices, with small laminated flash cards of the investigations available on the work station. The pathway will recommend that patients are reviewed, along with their results, within 1 month and appropriate next steps to be taken regarding patient management.

Usual care will also have the electronic template triggered but will only ask the patient for their consent to pass on contact details to the study team. The GPs and

clinicians in usual care will be asked to proceed with investigating the patient and their symptoms as per their usual practice and be directed to the NICE Clinical Knowledge summary for breathlessness[14] to standardise care.

## Outcomes

As this is a feasibility trial, a formal sample size/power calculation is not required. Recruitment itself is one of the main measures of feasibility. The feasibility measures are outlined in box 1 and the secondary outcome measures in table 1. Recruitment rate will be recorded as a proportion of participants consented compared with the number of participants identified as eligible by the GP practices.

## Data collection

Data will be collected in accordance with sponsor policies and standard operating procedures. Baseline data will be collected at research visit 1, within 1 month of the patient consulting their GP for breathlessness. Questionnaires will be completed by post at 6 months and visit 2 will occur at 12 months after visit 1. Healthcare records will be reviewed for all patients at 3 and 12 months to record the investigations performed and when, diagnosis and time to diagnosis.

Detailed healthcare utilisation data will be recorded at 12 months and 5 years, including hospital admissions, healthcare use and patient survival. This information will be collected from GP records and NHS digital.

## Anthropometric measures

BMI will be calculated by measuring the patient's height and weight. Body composition using bioelectrical impedance will provide measurements of lean mass and body fat percentage. Each participant's waist and hip circumference will be measured to the nearest 0.1 cm.

## Patient-reported outcome measures

### HRQoL

The Chronic Heart Questionnaire Self-Report is a validated and responsive questionnaire developed for patients with heart disease to assess HRQoL.[29] It has four domains: dyspnoea, fatigue, mastery and emotional function and a known minimal clinically important difference in patients with chronic heart and lung disease (Chronic Respiratory Questionnaire version). We will be using it in a population with undifferentiated breathlessness as part of feasibility.

The EuroQol 5 Dimension 5 Level questionnaire (EQ5D-5L)[30] will be used to assess generic HRQoL. The EQ5D-5L was chosen as it is a standardised measure of health status independent of disease used to calculate quality-adjusted life years (QALY).

### Breathlessness

The following questionnaires will be used to assess different aspects of breathlessness and to help select appropriate patient-reported outcome measures for the future trial:

1. Dyspnoea-12 is a brief 12 item, self-complete, questionnaire which has been found to reliably measure breathlessness in a variety of diseases.[23] Dyspnoea has both sensory and afferent components and this tool was developed to ensure both aspects could be measured.
2. The Multidimensional Dyspnoea Profile is a self-complete questionnaire for breathlessness divided into an immediate perception domain and an emotional response domain.[31] This questionnaire shows responsiveness to change in an acute and routine care setting for patients with breathlessness.[31]
3. The Baseline Dyspnoea Index and Transition Dyspnoea Index are short interviewer led questionnaires involving open questions about how their breathlessness affects everyday life.[32] This is measured over time in respect to what tasks a patient can manage and how much effort is required to complete a task.
4. The MRC Dyspnoea Grade 5-point scale is patient completed and requires participants to indicate to what extent their breathlessness limits their function by working down the statements which increase in severity regarding functional limitation.[33]

### Anxiety and depression

Participants will also complete the Hospital Anxiety and Depression scale which is a simple self-completed questionnaire with 14 questions relating to either anxiety or depression[34]

### Activation measure

The Patient Activation Measure is a self-completed questionnaire which assesses patients' knowledge, skills and confidence to manage their own health.[35]

### Physical activity

Daily physical activity and stationary time will be assessed for 7 days using the GT3x ActiGraph device that is worn around the waist,[36] and the wrist worn GENEActiv device.[37] Sedentary time, daily step count and time spent in moderate and vigorous physical activity will be recorded. The activity monitors will be fitted at visits 1 and 2 and will be programmed to begin collecting data at midnight. Data will then be collected for 7 days thereafter. To maximise the use of the data for comparison with other disease data sets we will use both devices as long as patients are willing. If participants can only use one device we will request that this is the wrist worn GENEActiv.

### Exercise capacity

This will be assessed using the Incremental Shuttle Walk Test (ISWT).[38] This walking test requires the patient to walk between two cones nine metres apart in time to a set of auditory beeps. Initially, the walking speed is very slow, but each minute the required walking speed progressively increases. The patient will walk for as long as they can until they are either too breathless or can no longer keep up with the beeps at which time the test ends. It is reported as the distance walked. The ISWT is found to

be valid, reliable and responsive in patients with chronic heart and respiratory disease.[39–41]

Heart rate, oxygen saturations, blood pressure and BORG breathlessness score will be recorded before and after completion of the walk tests. A practice walk test will be performed as per the guidelines for this exercise test.

### Frailty

Fried's frailty definition will be recorded, which is based on patient reported weight loss and exhaustion along with measured slowness (gait speed), weakness (hand grip) and physical activity.[42] This has been shown to provide a standardised definition of frailty with predictive validity in the community dwelling older population.[43] The Rockwood Frailty Scale will also be recorded. This is completed by the researcher in response to medical history and outcome measures taken during the visit and has also been demonstrated as a valid and reliable way of documenting frailty.[44]

Participants will also complete the Timed Up and Go test where the patient starts in a seated position, stands and walks 3 metres, then turns around and returns to the seated position.[45] The patient is timed how long this process takes.

The Short Physical Performance Battery[46] which includes the 4-Metre Gait Speed test and the Sit-to-Stand test and assessing balance with the patient standing in different positions (side by side stand, semi tandem stand, tandem stand) will be completed, along with handgrip and quadriceps strength.

### Health economic modelling

The feasibility study is structured to support the future RCT which will estimate the lifetime incremental cost per QALY gained. The objectives are to identify: the main NHS and prescribed specialist services cost components; the resource use and unit cost data required for each of these components; potential sources of HRQoL data suitable for estimating QALYs in this patient group; potential sources that could be used to estimate residual life expectancy and other long-term outcomes among patients.

### Semi-structured interviews

Audio-recorded interviews will be conducted privately face-to-face or via telephone between the participant and an interviewer, following informed consent. Interviews will take place with patients and GPs until data saturation is perceived. The interviews are anticipated to be between 30 min and 1 hour duration and will be professionally transcribed verbatim, with identifiable information removed. The transcription will be performed by an external company and a confidentiality agreement will be in place. Interview prompts will be devised based on relevant literature, experience of the team and consultation with patient representatives.

Patients consented for the feasibility trial who are willing and able, and healthcare staff from the participating practices, will be interviewed.

The interviews will explore patients' experience of breathlessness, taking part in the trial and their related healthcare. Patients will also be asked about the acceptability of the research visits and outcome measures performed and about their understanding of the trial. Interviews with healthcare staff will seek to understand what is usual care and any influence that taking part in the trial has on usual care. The clinician interviews will include questions about what a diagnostic pathway should or could look like from the perspective of the health professionals. They will also explore any barriers to screening patients for eligibility, challenges in implementing the pathway or perceived benefits of the intervention. The patient participant and clinician interview guides can be viewed in the online supplemental.

### Data management

Paper based anonymised study records will be stored in locked filing cabinets within a locked office at Glenfield Hospital. Electronic records will be stored on a restricted access, secure University of Leicester and University Hospitals of Leicester NHS Trust computer system, maintained by the Trust. Audio recordings will be done using an encrypted Dictaphone. The recordings will be uploaded to secure files on University of Leicester and University Hospitals of Leicester computers then deleted from the Dictaphone. Access to the files will be restricted and password protected.

### Data analysis

Data analysis will be performed in an exploratory fashion. Descriptive statistics, number and percentage for categorical data and mean and SD or median and IQR for non-normally distributed continuous data will be present for all demographics, baseline characteristics and questionnaire scores. Normality of the baseline characteristics will be determined using the Kolmogorov-Smirnov test or Shapiro-Wilk test depending on final recruitment numbers.

SPSS V.26 will be used for statistical analysis. GraphPad software will be used for any figures. Data analysis will be performed on the complete data set using all participants.

Secondary outcomes for both groups will be described as mean (SD) and median (IQR) for normally or non-normally distributed data, respectively. The time to diagnosis will be analysed using survival analyses based on Cox proportional hazards survival modelling. The proportion of patients with valid diagnosis at 3 months and 1 year will be described and compared using $\chi^2$ tests.

### Qualitative analysis

The interviews will be reviewed using thematic analysis,[47] supported by NVivo software. This approach follows six distinct stages: familiarisation with data; generating initial codes; searching for themes; reviewing themes; defining and naming themes and producing the report.[48] Initial coding will be carried out and a sample of interviews will be coded by a second member of the team to ensure

consistency and to enhance interpretive authenticity. Throughout the data analysis, an iterative approach will be undertaken with the research team meeting to discuss and review emerging themes and search for accounts that provide contesting views of the same phenomena or identify different phenomena. Analysis will continue until data saturation and themes will be synthesised and supported by using relevant quotes from the data. Patient representatives will be invited to comment on the emerging themes from the patient interviews to assess whether important issues may have been missed which could be included in subsequent interviews.

## Protocol amendments

Any changes to the study protocol outlined in this paper will be approved by Nottingham 1 Research Ethics Committee. This will be in agreement with Sponsor University of Leicester and University Hospitals of Leicester Research department.

## ETHICS AND DISSEMINATION
### Ethical approval

The Research Ethics Committee Nottingham 1 have provided ethical approval for this research study (REC Reference: 19/EM/0201).

### Monitoring

A Trial Steering Committee (TSC) will be convened to provide oversight and support to the project. The committee will comprise of an independent chair, independent members including clinicians, experts in breathlessness, statistician and policy experts, patient representative members and the principal investigator (PI). The trial coordinator will attend meetings as appropriate. A TSC charter will be put in place and 'Conflict of Interest' declarations obtained for all members and attendees. The TSC will meet as required to monitor the progress of the study, adherence to the protocol, progress of the study, consideration of new information of relevance to the research question and participant safety.

A Trial Management Group (TMG) has been established during the preparation of the study. Group members include the PI, research associate/project lead, trial coordinator and research assistants. Other collaborators and Leicester Clinical Trials Unit, specialties with specific expertise will attend as appropriate. The TMG will be held at least monthly to monitor all aspects of the conduct and progress of the study, ensure that the protocol is adhered to and take appropriate action to safeguard participants and the quality of the study itself.

### Dissemination

Results from the study will be disseminated by presentations at relevant meetings and conferences including British Thoracic Society and Primary Care Respiratory Society, as well as by peer-reviewed publications and through patient presentations and newsletters to patients, where available. The results will also be shared with local primary and secondary care partners. Following the feasibility trial, the aim is to conduct a national multicentre trial to assess clinical and cost-effectiveness of a diagnostic pathway for breathlessness. The feasibility outcomes collected and qualitative analysis will help refine the design of a future trial.

**Author affiliations**
[1]Respiratory Sciences, University of Leicester, Leicester, UK
[2]Leicester Clinical Trials Unit, University of Leicester, Leicester, UK
[3]Institute of Primary Care and Health Sciences, Keele University, Staffordshire, UK
[4]NIHR Leicester Biomedical Research Centre – Respiratory, University Hospitals of Leicester NHS Trust, Leicester, UK
[5]International Primary Care Respiratory Group, London, UK
[6]Barwell & Hollycroft Medical Centres, Leicester, UK
[7]Health Sciences, University of Leicester, Leicester, UK

**Acknowledgements** We acknowledge and are grateful to our public and patient involvement (PPI) members; Winifred Smart, Jagruti Lalseta, Brian Davies, Paul Ashby and Tony Watling. RAE thanks Professor Mike Morgan and the Research Design Service, University of Leicester for their support with the NIHR clinician scientist fellowship application. We also thank the clinicians who contributed to the development of the structured diagnostic pathway including Professor Ruth Green, Mrs Louise Clayton, Mrs Karen Moore, Mr Alex Woodward, Mrs Jo Szymkowiak, Ms Alison Scott and Ms Jane Giles. The work is supported by the NIHR Leicester Biomedical Research Centre - Respiratory.

**Contributors** RAE conceived the research idea, and developed the theory and plan for this study. GD and RAE drafted the initial manuscript. SW developed the electronic template for use in the recruitment strategy. All authors (RAE, GD, JC, SW, SC, SE, NB, DJ, NA and MS) contributed to the study development and reviewed, commented and approved the manuscript.

**Funding** This work was funded by a NIHR Clinician Scientist Fellowship (CS-2016-16-020) awarded to Dr RAE. NA is supported by a Health Foundation Improvement Science Fellowship and also by the National Institute for Health Research (NIHR) Applied Research Collaboration East Midlands (ARC EM).

**Disclaimer** The views expressed are those of the authors and not necessarily those of the National Health Service, the NIHR, or the Department of Health.

**Competing interests** None declared.

**Patient consent for publication** Not applicable.

**Provenance and peer review** Not commissioned; externally peer reviewed.

**ORCID iDs**
Gillian Doe http://orcid.org/0000-0003-4782-5811
Rachael A Evans http://orcid.org/0000-0002-1667-868X

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
