## [Reviewer comments · BMJ Open]

ARTICLE DETAILS

TITLE (PROVISIONAL)	A protocol for a feasibility study of a multi-centre cluster randomised control trial to investigate the clinical and cost effectiveness of a structured diagnostic pathway in primary care for chronic breathlessness.
AUTHORS	Doe, Gillian; Clanchy, Jill; Wathall, Simon; Chantrell, Stacey; Edwards, Sarah; Baxter, Noel; Jackson, Darren; Armstrong, Natalie; Steiner, Michael; Evans, Rachael A

VERSION 1 – REVIEW

REVIEWER	Yet Hong Khor Austin Health, Respiratory and Sleep Medicine
REVIEW RETURNED	25-Sep-2021

GENERAL COMMENTS	This manuscript describes the protocol of a feasibility study of multi-centre RCT of the efficacy and cost-effectiveness of a structured diagnostic pathway in primary care for the management of adults presenting with chronic breathlessness. The study rationale is sound to address this much-needed area. The study design is appropriate and well-considered. The manuscript is well written. A few comments to improve clarity: 1. Consider shortening the manuscript title to be more succinct.2. Table 1: Should include data collected from interviews with GP regarding their experience in the trial3. Is there a pre-specified timing for participant recruitment, e.g. only during the first consultation?4. How will the structured diagnostic pathway be provided to GPs and clinicians, e.g. online as an automatic prompt during the consultation, paper-based, or both?5. It is unclear whether a preliminary health economic modelling will be performed in this feasibility study. Health economic modelling has been listed in the Methods, although not included in the Data Analysis.6. In view of the aim (4) to identify sources for economic evaluation, will there be data collection of healthcare utilisation (e.g. hospitalisation, ED presentation, etc)?7. Page 17, Line 52: Suggest rephrasing this sentence to improve clarity: "The feasibility study is based on the future RCT estimating the lifetime incremental cost per QALY gained". E.g. "The feasibility study is structured to support the future..."8. The topic guide (interview prompts) for the semi-structured interviews should be provided, at least as a supplement.9. Given the current environment with COVID-19, is there a contingency plan to allow minimal interruption of this study if there is a local surge? E.g. will telehealth be used?
--

VERSION 1 – AUTHOR RESPONSE

Reviewer Report:

(Reviewer: 1, Dr. Yet Hong Khor, Austin Health)

Comments to the Author:

This manuscript describes the protocol of a feasibility study of multi-centre RCT of the efficacy and cost-effectiveness of a structured diagnostic pathway in primary care for the management of adults presenting with chronic breathlessness. The study rationale is sound to address this much-needed area. The study design is appropriate and well-considered. The manuscript is well written. A few comments to improve clarity:

Authors response:

Many thanks for your time and positive comments.

1. Consider shortening the manuscript title to be more succinct.

Authors response:

After careful consideration, we have shortened the title as follows:

A protocol for a feasibility study of a multi-centre cluster randomised control trial to investigate the clinical and cost effectiveness of a structured diagnostic pathway in primary care for chronic breathlessness.

2. Table 1: Should include data collected from interviews with GP regarding their experience in the trial

Authors response:

Thank you for this helpful suggestion to add the qualitative data to Table 1. This has been added (page 6).

3. Is there a pre-specified timing for participant recruitment, e.g. only during the first consultation?

Authors response:

Participants can be recruited if they are within two presentations to the GP for their breathlessness. This is included on page 8 under Eligibility Criteria:

Patients will be eligible if they are over 40 years old, experienced breathlessness for over two months, and are within their first two presentations to primary care with symptoms of breathlessness.

4. How will the structured diagnostic pathway be provided to GPs and clinicians, e.g. online as an automatic prompt during the consultation, paper-based, or both?

Authors response:

We have clarified this important point on page 11. The new text additions are in red below:
The electronic template will prompt the GP to action these investigations. The pathway will also be provided as a laminated document for each clinician in the Intervention practices, with small laminated flash cards of the investigations available on the work station.

5. It is unclear whether a preliminary health economic modelling will be performed in this feasibility study. Health economic modelling has been listed in the Methods, although not included in the Data Analysis.

Authors response:

Thank you for this query. We will not be conducting health economic modelling as part of the feasibility study. We aim to identify and collect data to inform the economic evaluation in a future larger RCT. We have included this in one of the feasibility aims on page 6. We have also altered the title in the methods to clarify:

Health Economic modelling for the future trial

6. In view of the aim (4) to identify sources for economic evaluation, will there be data collection of healthcare utilisation (e.g. hospitalisation, ED presentation, etc)?

Authors response:

This is described on page 12 under Data Collection:

Detailed health-care utilisation data will be recorded at 12 months and five years, including hospital admissions, healthcare use, and patient survival. This information will be collected from GP records and NHS digital.

7. Page 17, Line 52: Suggest rephrasing this sentence to improve clarity: “The feasibility study is based on the future RCT estimating the lifetime incremental cost per QALY gained”. E.g. “The feasibility study is structured to support the future...”

Authors response:

Thank you for your comment and suggested rephrasing. We agree this helps clarify and now reads as follows:

The feasibility study is structured to support the future RCT which will estimate the lifetime incremental cost per QALY gained.

8. The topic guide (interview prompts) for the semi-structured interviews should be provided, at least as a supplement.

Authors response:

Thank you for this helpful and valid suggestion. We have added as a supplementary file and referenced on page 16:

The patient participant and clinician interview guides can be viewed in the online supplement.

9. Given the current environment with COVID-19, is there a contingency plan to allow minimal interruption of this study if there is a local surge? E.g. will telehealth be used?

Authors response:

Thank you for addressing this important point. The involvement from the GP practices will continue as outlined in the protocol with the electronic prompt being utilised and investigations completed, even where the initial GP consultation has taken place by phone.

We have submitted an amendment to include telephone consent and remote participation by phone and post where patients would prefer not to attend a face to face research appointment.

VERSION 2 – REVIEW

REVIEWER	Yet Hong Khor Austin Health, Respiratory and Sleep Medicine
REVIEW RETURNED	15-Oct-2021
GENERAL COMMENTS	All comments have been addressed adequately.